# *Beauveria bassiana* Endophytic Strain as Plant Growth Promoter: The Case of the Grape Vine *Vitis vinifera*

**DOI:** 10.3390/jof7020142

**Published:** 2021-02-16

**Authors:** Spiridon Mantzoukas, Ioannis Lagogiannis, Dionusia Mpousia, Aristeidis Ntoukas, Katerina Karmakolia, Panagiotis A. Eliopoulos, Konstantinos Poulas

**Affiliations:** 1Department of Pharmacy, University of Patras, 26504 Patras, Greece; lagoipp@gmail.com (I.L.); mpoudionusia95@gmail.com (D.M.); ntoukasaris7@gmail.com (A.N.); 2ELGO-Demeter, Plant Protection Division of Patras, 26444 Patras, Greece; aikkair@gmail.com; 3Department of Agrotechnology, University of Thessaly, Gaiopolis, 41500 Larissa, Greece; eliopoulos@uth.gr

**Keywords:** *Beauveria bassiana*, endophytic, growth parameters, *Tribolium confusum*, *Vitis vinifera*

## Abstract

The common grape vine, *Vitis vinifera*, is a widely known plant with commercial and pharmacological value. The plant hosts a variety of microorganisms known as endophytes that can live within the tissues of the plant for a considerable time period, or even their whole life cycle. The fungus *Beauveria bassiana* is a well-studied endophyte which can colonize a variety of plants in many ways and in different parts of the plant. In this study, we examined the effect of the endophytic fungus *B. bassiana* on the growth of *V. vinifera*. The results demonstrated not only a successful colonization of the endophyte, but also a noteworthy impact on the growth of the *V. vinifera* root without harming the plant in any way. The fungus was also re-isolated from the parts of the plant using inst bait method. Overall, the study demonstrates the capability of *B. bassiana* to colonize *V. vinifera* plants, adding to the already existing knowledge of its endophytic activity, and highlighting its beneficial impact on the root growth.

## 1. Introduction

*Vitis vinifera* (Rhamnales: Vitaceae) L. (grapevine) is widely cultivated globally and is a native fruit to Europe and Western Asia. Besides its commercial value, the plant exhibits important pharmacological effects such as antioxidant, antimicrobial and cardioprotective properties, that can be attributed to its bioactive compounds (flavonoids, polyphenols, anthocyanins and more) [1,2]. *V. vinifera* hosts numerous microbial organisms, which have a positive impact on the quality of the wine, and both endophyte and host benefit from this interaction [3].

Endophytes constitute a relatively underexplored and attractive source of natural products suitable to be exploited in medicine, agriculture or industry [4,5,6,7,8,9]. The category of endophytes comprises several microorganisms such as bacteria and fungi, whose existence is plant-associated as they inhabit the interior of living plant tissues for most, if not all, of their lives. Endophytes can be introduced into the plant by means of root drench, foliar spray and/or seed treatment [10]; they shield themselves from other microorganisms by producing several metabolites (e.g., alkaloids, flavonoids, terpenoids) which have also been found to protect the plant against potential pests [11,12,13,14,15]. In fact, endophytic fungi are characterized by a high diversity of secondary metabolites which render them a source of considerable biotechnological potential [16,17]. Moreover, some endophytes can trigger plant defense genes and systemic self-protection mechanisms against the attack of certain pathogens [18]. They also can contain the action of plant pathogenic fungi by competing with them for space and nutrient uptake [11,12,13,14,15,18].

Plant-associated microorganisms are also known to enhance host adaptation to environmental changes [19,20]. Endophytes can presensitize the cell metabolism of plants in a way that the primed plants can address and survive environmental stress more efficiently than non-primed plants [21,22]. Finally, throughout their symbiosis, the metabolic pathways of host and endophyte are interlinked via the exchange of compounds which enable proper metabolic functioning [23]. Overall, increase of nutrient availability, the production of antibiotics and plant growth enhancing substances and the improvement of soil quality and the induction of plant defense mechanisms are some of the ways in which endophytes can support plant growth [24,25,26].

A major result of the endophytic activity of these microorganisms lies in how they affect the development and the morphology of the plant [27]. Symbioses of plants with endophytic organisms have been shown to increase plant growth rate [28,29], strength, and capability in nutrient uptake [30]. Previously published research refers to an enhancement of plant morphological features such as height, weight, dry biomass, and root characteristics [31]. An additional aspect of this symbiosis lies in the aid that is provided to the plants in their attempt to survive and develop in challenging conditions [32,33].

One of the most known endophytes is the entomopathogenic fungus *Beauveria bassiana*, (Hypocreales: Cordycipitaceae) (Bals.-Criv.) Vuill, which is considered one of the most abundant species globally. Its major characteristic is that this fungus is conferred natural antimicrobial and insticidal properties by the substances that it produces [10,34]. Some of the most important mechanisms for insects control include pathogenicity, antagonism, systemic resistance, and the tritrophic action associated with natural enemies [10,11,12,13,14,15,16,34,35,36,37,38,39,40]. This feature makes it a desirable product in agriculture, because of its easy application and natural profile. It can be delivered in multiple ways, mainly using conidial suspensions. The form and method of its application as well as the part of the plant that will be targeted depends on the plant species [10,35].

*B. bassiana* is able to colonize a variety of plants (e.g., rice, onion, tomato, palm tree, grape, wheat, maize, coffee, cacao, pine, pumpkin, cassava, bean, potato, and cotton) [36,37,38,39,40,41,42,43,44,45,46,47,48]. It mainly occurs in roots, internal tissues, foliage, and stems of the plant [35,49]. It is believed to provide its host with nitrogen derived from the infested insects; in return, it receives carbon-rich substances from the host [50]. Research has been widely conducted regarding the impact of this fungus on the development of the organism it colonizes. It has been established that a series of plant growth parameters such as dry biomass and the development of tissues are enhanced by its presence [43,50,51,52].

The aim of this study is to clarify and highlight the impact of a local endophytic *B. bassiana* strain on *V. vinifera* growth. Our study focuses not only on the on the colonization rate of the endophyte in various parts of the plant, but also on the development of leaves, shoots, stems, and roots. 

## 2. Materials and Methods

### 2.1. Plant Material, Fungus, Inst and Experimental Design

*V. vinifera* cuttings (var Sideritis R110 and var Sideritis self-rooted) were planted into 20-l clay pots filled with a sand:peat mixture (1:2 *v/v*) and transferred to an open area near the EMBIA Laboratory (Department of Pharmacy, University of Patras, Greece; 38.2° N, 21.5° E) where they grew under full sunlight, receiving 800 mL of water per plant, per day. Thirty days after plantation, 200 plants (closely similar in terms of plant height and number of leaves) were randomly assigned to two groups: one group consisting of two subgroups, var Sideritis rootstock R110 (50 plants) and var Sideritis self-rooted (50 plants), was assigned for inoculation with the fungus. The other group which also contained two subgroups, var Sideritis R110 (50 plants) and var Sideritis self-rooted (50 plants), served as the control. Measurements for plant physiology were made on days 0, 7, 14, 21, and 53 of the experiment. Day 0 marked the initiation of the experiment. Subsequently, treated plants were irrigated once and control plants were irrigated with ddH_2_O (double distilled water). The irrigation procedure was performed late in the afternoon (at 25–28 °C), and the soil was covered with aluminum foil for 48 h to maintain its humidity. The experiment was set up as a randomized block design in a 4 × 2 factorial design involving factor of treatment (1: *B. bassiana*; 2: ddH_2_O) by and factor B (the cultivars, var Sideritis rootstock R110 and var Sideritis self-rooted) (Figure 1).

The fungal isolate *B. bassiana* H2S32 was used to inoculate plants. The isolate was selected because it had been isolated from a *V. vinifera* var. *Sideritis* plantation using the *Tribolium confusum* (Coleoptera: Tenebrionidae) Jacquelin du Val bait method [53]. More specifically, soil samples had been collected from the vine terroirs of the Krokidas Winery in the prefecture of Achaia [54]. Fresh conidia were collected from the SDA cultures after 15 days and transferred to a 500 mL glass beaker with 100 mL sterile distilled water containing 0.05% Tergitol NP9. The conidial suspension was filtered across 10–12 layers of sterile cloth to remove hyphal debris, then homogenized by mixing on a magnetic stirrer for 5 min [55]. The fungus was grown in 9 cm Petri dishes with Sabouraud Dextrose Agar (SDA; Sigma-Aldrich, Seelze, Germany) in the dark for 15 days, at 25 ± 2 °C. The concentration of conidia was adjusted to 10^8^ conidia mL^−1^ [56] using a Neubauer haemocytometer under a phase contrast microscope at 400× magnification (Axioplan; Zeiss, Oberkochen, Germany). The assessment of conidia viability pointed to a germination exceeding 95%. This was established by examining conidia at 40× magnification after they had been incubated for 24 h on SDA.

*T. confusum* was reared on whole wheat flour. We selected it as an inst bait for the isolation of the fungus, based on its considerably good performance in previous studies [57,58]. The adult insts were kept in cages at 25 ± 2 °C, relative humidity 60–70%, and a photoperiod 16:8 h (light-darkness) (PHC Europe/Sanyo/Panasonic Biomedical MLR-352-PE) (Department of Pharmacy, University of Patras, Patra).The initial population was provided in 2018 by the University of Thessaly, Department of Agrotechnology, Crop Protection Lab (Larisa, Greece), and it has since then been continuously reared in the Laboratory of Molecular Biology and Immunology, Department of Pharmacy, University of Patras.

### 2.2. Endophytic re Isolation from Plant Tissues

Eight samples of *V. vinifera* leaves, stems, shoots, and roots were cut into 1-cm^2^ diameter and 0.5 cm^2^ thick discs in a laminar flow chamber. Measurements for endophytic leaf colonization were made on days 7, 14, 21, and 53 of the experiment, and of stems, shoots, and roots on day 53 (end of the experiment). The samples were surface sterilized by immersion in 96% ethanol solution for one minute, in 6% sodium hypochlorite solution for five minutes and, finally, in 96% ethanol solution for thirty seconds [59]. Sterile leaves, stems, shoots, and roots samples were then inoculated SDA substrate using a sterile metal hook, then incubated in the dark at 25 °C ± 2 and 80% humidity. The conidial growth sequence lasted 14 days at sealed petri with parafilm.

The germination of fungal conidia on the *V. vinifera* leaves, stems, shoots, and roots was evaluated using an optical microscope (40×). The number of samples which displayed fungal growth was calculated using the following formula: number of *V. vinifera* samples with fungal growth/total number of samples [57]. The above-mentioned process was completed inside a laminar flow chamber (Equip Vertical Air Laminar Flow Cabinet Clean Bench, Mechanical Application LTD, Athens, Greece) in the Laboratory of Molecular Biology and Immunology, Department of Pharmacy, University of Patras.

### 2.3. Effect of Endophytic B. bassiana on V. vinifera Plants

At the end of the experiment, the number of leaves per plant, the thickness of the stem per plant measured with caliper, the length of clematis per plant, and the height of plants (as the distance from the apical stem point to the soil) were measured (*n* = 50); plants from each treatment block were then uprooted. Each plant was divided into leaves, stems, shoot and root. Leaves, shoot, and stems were further cut into smaller pieces, while roots were thoroughly washed of any soil particles; all tissues were finally oven dried at 80 °C for 72 h. The dry mass of leaves, shoots, stems, and roots was recorded, and we also estimated the ratio of the aboveground to the belowground biomass (*n* = 100). Aboveground biomass is the sum of leaf and stem biomass, belowground biomass is the root biomass.

### 2.4. Fungi Re-Isolation from the Soil

Ten soil samples (10 g/sample) were taken from each clay pot. The collected soil was placed in small plastic containers (2 cm × 3 cm). Afterwards, it was sieved (metal sieve, 2 mm × 1 mm, Aggelis Equipment, Athens, Greece) and placed in Petri plates. Then, (10) newly emerged adults of *T. confusum* beetle were placed in Petri plates with the 10 g of the soil from the clay pots. The Petri plates were stored at room temperature (25 ± 2 °C) for 18–20 days, and for the first 4 days, the plates were inverted daily so that the adults could move throughout the soil samples. Mortality control was monitored at 2, 4, and 6 days, respectively. Dead adults were immersed in 6% NaOCl_2_ for 3 s, in order to be sterilized (to avoid developing saprophytic fungi). They were then placed in sterilized Petri plates with a No. 1 Whatman paper impregnated with ddH_2_O (double distilled water) until mycelia appeared. *T. confusum* cadavers showing external mycelial growth were identified by examining each cadaver using a Stemi 2000 stereomicroscope (Carl Zeiss^®^, Jena, Germany). These samples were kept in the dark at 25 ± 1 °C and observed using a stereoscope to determine possible infection by *B. bassiana*. Daily conidia from the infected adults were placed in 9 cm sterilized Petri plates on a layer of SDA, for the isolation of entomopathogenic fungi. The plates were kept at 25 ± 1 °C in the dark to achieve the incubation and development of the fungi. When a fungus was developed, it was re-isolated to acquire a pure culture. The above-mentioned process was carried out inside a laminar flow chamber (Equip Vertical Air Laminar Flow Cabinet Clean Bench, Mechanical Application LTD). 

To estimate the number of fungal colonies surviving in the soil, one gram of soil was weighed out of each sample, to which 9 mL of ddH_2_O with 0.05 Triton X-100 was added. The resulting solution was intensely shaken for 30–40 s. Following this, 0.1 mL of the soil solution was spread out on a selective medium with a glass spatula. The selective medium consisted of 1 L of water, 20 g of glucose, 18 g of agar, and 10 g of peptone [60,61]. After sterilization and cooling of the agar, the following selective components were added to the medium: 0.6 g of streptomycin sulfate, 0.005 g of chlortetracycline, 0.05 g of cyclo-heximide, and 0.1 g of dodine. The experiment randomized block design was performed in ten Petri dishes per sample. The dishes were kept at 25 ± 1 °C in the dark in order to facilitate the incubation and development of fungi. After 14 days, the colonies of the fungus were counted. The results were expressed as the number of colony-forming units (CFU) of fungi in 1 g of soil.
[CFU g^−1^soil] = 2 × colonies per plate/dilution × 10^3^(1)

### 2.5. Fungi Identification Methods

The isolates after were subcultured several times on plates with Sabouraud Dextrose Agar (SDA) to ensure purity and monosporic cultures then morphologically identified using ZEISS Primo Star microscope (Carl Zeiss Microscopy GmbH, Germany) at 400× magnifications. The selected isolates were stored in the microorganism’s repository of the EMBIA Laboratory, Department of Pharmacy, School of Health Sciences, University of Patras. The conidia were scraped from the surface of the plant tissues and from the dead cadavers using a sterile loop and transferring the conidia to Potato Dextrose Agar (in-house technique). Following the method outlined by Rogers and Bendich [62], the genomic DNA (gDNA) was extracted. Applying universal primer sets ITS4 (5′-TCCTCCGCTTATTGATATGC-3′) and ITS5 (5′-GGAAGTAAAAGTCGTAAC AAGG-3′), a fragment of the ITS spacer region was expanded. PCR reactions (30 µL) included 50 ng of template gDNA, 1.25 µL of each 10 μM oligonucleotide, 1 µL of 10 mM dNTPs, 1 µL of 2 U/µL Taq DNA polymerase (Minotech), 1.5 μL of MgCl_2_, 2.5 µL of 10× PCR buffer. The PCR protocol for amplification of ITS regions includes 31 cycles, at 94 °C for 60 s, 55 °C for 60 s, and 72 °C for 90 s, followed by a final elongation at 72 °C for 5 min. PCR products were kept at 4 °C. The quantity and quality of PCR products were determined by gel electrophoresis using 2% agarose gel, which was stained with SYBR Safe DNA Gel Stain (Invitrogen) and visualized under UV light (BIO-RAD, Molecular Imager Gel Doc XR System). The amplified products were purified and sequenced in the laboratory of CeMIA SA (Company for Cellular and Molecular Immunological Applications), University of Thessaly. DNA sequences collected from this work were matched with the Basic Local Alignment Search Tool (NCBI BLAST) [63].

### 2.6. Statistical Analysis

All statistical analyses were conducted using the SPSS v.25 (IBM-SPSS Statistics, Armonk, NY, USA). For growth measurements, colonization percentage, *T. confusum* mortality and mycelium on dead cadavers, one-way ANOVA was performed. Bonferroni’s post-hoc test was used to compare means of treatments. 

## 3. Results

### 3.1. Molecular and Microscopic Identification of the Fungi

Observations of cadavers indicated that the external mycelium appears on them within 72 h after their placement on damp filter paper. B. bassiana fungi were morpho-logically and molecular identified on plant tissues and on the dead cadavers (Table 1 and Figure 2).

### 3.2. Re-Isolation of Entomopathogenic Fungi from Leaves, Shoots, Stem and Roots on SDA Substrate

The establishment of endophytes in leaves of *V. vinifera* plants was evaluated at 7, 14, 21, and 53 days after inoculation (Figure 3). The colonization of shoots, stem, and roots was evaluated 53 days after inoculation. Successful re-isolations of the fungus were obtained from the leaves, shoots, stem, and roots of inoculated plants. A decline in leaf colonization by *B. bassiana* was observed after 53 days (21 % SR –24 % R110) (F = 2.712, df = 1, *p* = 0.352). Stem (F = 0.054, df = 1, *p* = 0.579) and shoot (F = 0.245, df = 1, *p* = 0.414) colonization was lower compared to leaf colonization after 53 days. On the other hand, root colonization was higher to that of leaves after 53 days (F = 1.154, df = 1 *p* = 0.279).

### 3.3. Growth and Mass Parameters

Growth and dry mass parameters measured at final harvest are depicted in Table 2 and Figure 4. The endophyte *B. bassiana* affected the leaf number (F = 2.860, df = 3, *p* = 0.001), height of plants (F = 2.377, df = 3.575, *p* = 0.38), length of root (F = 4.198, df = 3, *p* = 0.001) and the dry mass of roots (F = 5.811, df = 3.690, *p* = 0.001). On the other hand, it did not affect the diameter of the stem (F = 0.198, df = 3, *p* = 0.905), the length of the clematis (F = 0.927, df = 3, *p* = 0.315), the dry mass of leaves (F = 0.379, df = 3, *p* = 0.778), the dry mass of shoot (F = 0.585, df = 3, *p* = 0.667), and the dry mass of stem (F = 0.411, df = 3, *p* = 0.793). The aboveground-to-belowground biomass ratio showed an upward tendency (F = 9.111, df = 3, *p* = 0.001).

### 3.4. Fungus Soil Re-Isolation with T. confusum Bait

The mortality percentage *T. confusum* bait was significantly different (F = 35.365, df = 3, *p* = 0.001) (Table 3). The highest mortality was recorded in the soil collected from the self-rooted plant variety inoculated with *B. bassiana* (Table 3). The mortality of *T. confusum* adults was between 44.06 (2d) and 91.09 (6d) % in the soil from the self-rooted inoculated variety, and between 52.79 (2d) and 87.47(6d) % in the soil from the R110 inoculated variety. The control mortality was between 3.6 (2d) and 7.89 (6d) % in the soil from the self-rooted inoculated variety, and between 4.54 (2d) and 8.32 (6d) % in the soil from the R110 inoculated variety. The mycelium percentage was significantly different, at 87.97% of dead *T. confusum* adults in the soil from the self-rooted inoculated variety, and at 73.45% of dead adults in the soil from the R110 inoculated variety (Table 3) (F = 12.312, df = 3, *p* = 0,001). In the soil samples, fungi formed 29.1 × 10^7^ g^−1^ CFU in 1 g of soil from the self-rooted inoculated plants, and 17.8 × 10^6^ g^−1^ CFU in 1 g of soil from the R110 inoculated plants.

## 4. Discussion

Several studies have investigated microbial communities in the soil and their effect on plants [14,57,61,64]. Colonization of internal plant tissues of many crops has been achieved with *B. bassiana* [41,53,61], suggesting that these entomopathogens have the potential to colonize many different plant species. In previous studies, the *B. bassiana* strain GHA was used as an endophyte for *V. vinifera* plants [14,30]. In our study, we tested the native strain of *B. bassiana* H2S32 an endophyte, which was isolated from the soil of an organic *V. vinifera* cultivation in Achaia, Greece, using *T. confusum* as bait [54]. To deem an experiment of this type successful, three criteria must be met: colonization of plants by the endophytes, symptomless physiology of the endophyte plants, and a positive effect on the growth of the plant after colonization. The present study shows for the first-time that a native strain of endophytic *B. bassiana* can successfully colonize *V. vinifera* rootstock R110 and self-rooted varieties. The inoculation method used here resulted in colonization by *B. bassiana* for up to eight weeks after treatment and the results demonstrated that endophytic colonization was achieved in many plant tissues and especially in the actively growing roots of *V. vinifera*. This suggests that successful endophytic colonization by *B. bassiana* can be achieved in all plant tissues and especially in the actively growing roots of *V. vinifera*. 

The irrigation method was selected for the endophytic investigation of *B. bassiana* in *V. vinifera* plants. Considering that the inoculation method is crucial for the efficiency of the colonization of the tested plant [48], we opted for irrigation to enable the dispersal of conidia in the soil, the roots and within the plant. In the current study, colonization differed among various plant parts. We re-isolated *B. bassiana* from leaves (7 d, 14 d, 21 d and 53 d), roots (53 d), stem (53 d), and shoots (53 d). The colonization of the different plant parts confirms the transfer of the fungus within the plant system. It was high in leaves, stem, and especially in roots. The reason for the higher colonization of the roots is not clear, but it could be due to the inoculation method, which also reflects differences in the microbial and physiological conditions in the different plant parts. Many endophytic fungi show a certain degree of tissue specificity because they are adapted to conditions present in each organ [65,66,67]. Petrini and Fisher [68] and Liang-Dong et al. [69] reported that endophytic fungi exhibited tissue specificity because they are adapted to conditions present in each plant part. Allegrucci et al. [70] showed that tomato plants can be colonized by application of fungal conidia at the leaves. Greenfield et al. [71] displayed higher colonization rates by *B. bassiana* from the root of cassava plants across all their experiments. *B. bassiana* can colonized differently among plants parts such roots, leaves, and stems [46].

In cultivated grapevine, the composition of fungal communities is significantly associated with the rootstock genotype and shows different interactions; the nature of the rootstock has been addressed in our study as a possible variable explaining endophyte diversity. Marasco et al. [72] investigated how different rootstocks could affect the recruitment of bacteria from the surrounding soil and found that bacterial community diversity and networking in the roots were profoundly influenced by the rootstock type. We isolated *B. bassiana* from surface sterilized plants tissues of *V. vinifera*. This indicates that the fungi were systemic within the plants and remained localized in high CFU values in the soil, which includes the roots. The interaction between endophytes and plants occurs in different areas, including the root and plant surfaces. Regarding the provenance of the endophytic microbiota and its way of entry into plants, it is generally assumed that the main route is from soil to roots with endophytes ascending to the epigeal parts or alternatively through gaps along the plant overall surface, including wounds or stomata [73]. Our results show that the conidia of *B. bassiana* were concentrated in the upper soil strata, where the proximal end of the root is located. This systematic presence of the fungi agrees with other studies that have concluded that *B. bassiana* can establish itself as an endophyte throughout the entire plant, particularly after seed inoculation [36,74,75,76]. The high presence of the fungus in the soil, based on mortality data, the mycelium percentage on the dead *T. confusum* baits, and the increased number of CFUs, could be accounted for by the high vitality of *B. bassiana* in several temperatures and soil water content conditions [77]. In addition, *B. bassiana* can continue to grow in the saprophagous phase [78,79]; the pathogenicity of entomopathogenic fungi is a complex phenomenon whereby several factors other than the number of CFUs (e.g., cuticle degrading enzymes, strain genetic variation, gene expressions) can add to the virulence of the fungus over the inst.

Effects of established endophytes should be neutral for the plant in case of physiology and growth [58,80]. Occasionally, under stressful conditions, endophytes may be favorable for their hosts, enhancing their resistance against adverse environmental factors [81], such as drought and nutrient depletion [30,82], or strengthening their defense against biotic stresses such as herbivores and pathogens [14,46,55,61,64,83]. The presence of the endophytic *B. bassiana* used in this study neither suppressed *V. vinifera* growth nor reduced dry mass until final harvest. In some cases, the presence of the endophytic *B. bassiana* acted positively as the root dry weight results show in this study.

We cannot know whether systemic endophytic colonization of *V. vinifera* by entomopathogens would be successful in protecting the plant against pests of the leaves, such as European grapevine moth which represents a major pest in Greece. The mechanisms involved in the control of arthropod pests and diseases using endophytes include an interplay of factors such as antagonism, induction of plant host defenses, and host plant tolerance [39,84,85]. However, an endophytic establishment of entomopathogenic fungi inside *V. vinifera* plants could overcome problems commonly encountered when fungal-based products are sprayed directly onto leaf surfaces, such as poor persistence of fungal spores in the environment or high susceptibility to environmental factors like UV radiation or heavy rainfall [86]. On the other hand, some strains of entomopathogenic fungi are known to produce metabolites which might enter the food chain if the fungus occurs as an endophyte [87], an aspect which is an important issue for the registration process of fungal-based products. Nevertheless, entomopathogenic fungi like *B. bassiana* have shown to express multiple roles in protecting plants from insect pests, which is particularly relevant for the design of efficient management strategies against *V. vinifera* pests.

## 5. Conclusions

The significant endophytic colonization of *V. vinifera* by *B. bassiana,* and its considerable soil presence suggest that this isolate is well adapted to a wide range of conditions, rendering it endophytic in plants and pathogenic to insts. Drawing on our results, *B. bassiana* appears to become established as an endophyte in the *V. vinifera* without adversely affecting the plant. The fact that the development of the root increases by the endophyte *B. bassiana* might affect the survival rate of the plant, given the fact that the plant can get higher stability and can reach nutrients easier. This study provides the basis for further investigations, which should focus on the response of different *V. vinifera* varieties to different strains of *B. bassiana*, the latter’s long-term establishment throughout the entire life of the inoculated plants, as well as the virulence of the endophytic fungi against important pests of *V. vinifera.* Future studies could evaluate the presence of fungal entomopathogens inoculated into the soil in the different soil strata to determine if conidia adhere to soil particles in the upper soil layer around plant roots. 

## Figures and Tables

**Figure 1 jof-07-00142-f001:**
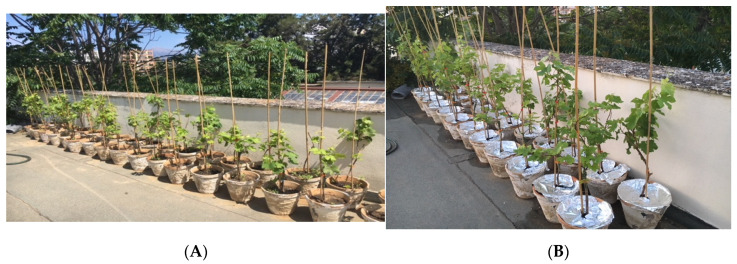
Experimental randomized block in clay pots filled with a sand: peat mixture at the open area near the EMBIA Laboratory. (**A**) *V. vinifera* plants before inoculation and (**B**) *V. vinifera* plants after inoculation.

**Figure 2 jof-07-00142-f002:**
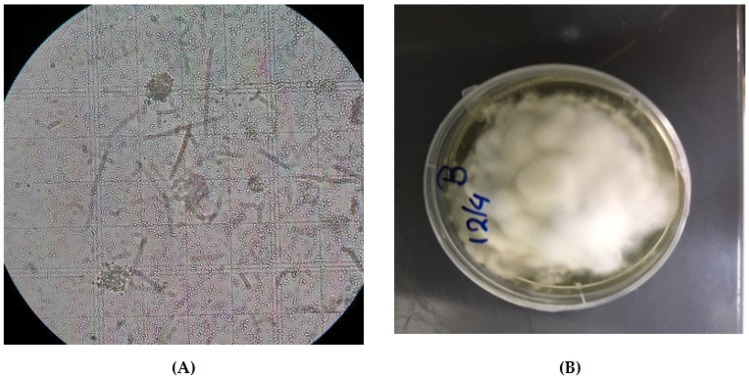
(**A**) Morphological identification of *B. bassiana* H2S32 from plant tissues and dead cadavers under ZEISS Primo Star (Carl Zeiss Microscopy GmbH, Germany) at 400× magnification, (**B**) *B. bassiana* H2S32 on plates with Sabouraud Dextrose Agar (SDA).

**Figure 3 jof-07-00142-f003:**
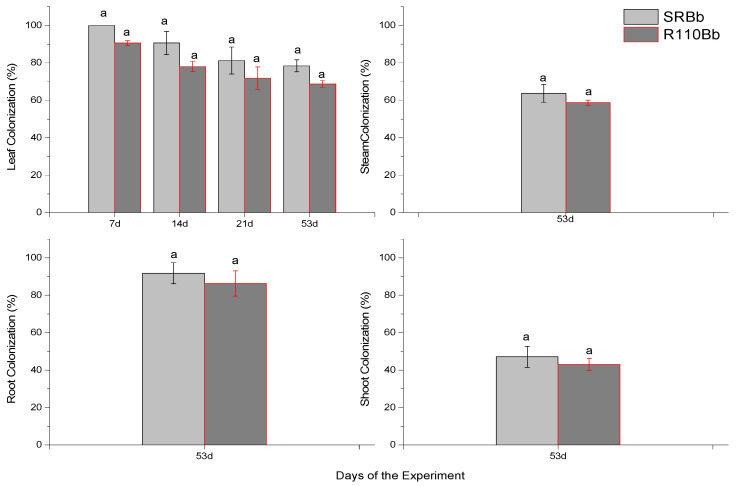
Mean (±sd; *n* = 50) colonization of *V. vinifera* leaf parts by *B. bassiana*, at 7 days, 14 days, 21 days, and 53 days after irrigation. Mean (±sd; *n* = 50) colonization of Stem, Shoot and Roots of *V. vinifera* by *B. bassiana*, at 53 days after irrigation. Mean ± sd values of different plant groups (SR or R110) with the same letter are not significantly different (Bonferroni’s test: *p* < 0.05). SRBb: self-rooted grapevines infested with *B.bassiana*, R110Bb: R110 rootstocks grapevines infested with *B.bassiana*.

**Figure 4 jof-07-00142-f004:**
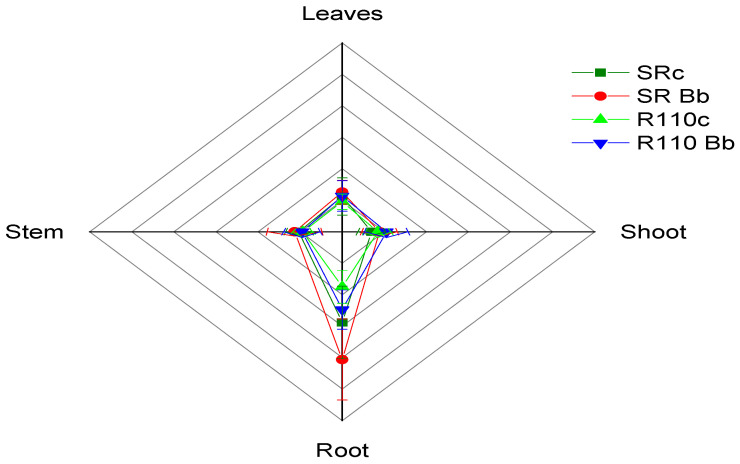
Two-dimensional presentation of dry weight of plant parts variables (Stem, Leaves, Shoot, Root) of endophyte-free *V. vinifera* plants (control) and plants infested with *B. bassiana* (OriginPro 9.0).

**Table 1 jof-07-00142-t001:** Molecular and Microscopic Identification from plant samples and *T. confusum* dead cadavers Blast ID Number, DNA, Sequence length, Conidia microscope evaluation, and Match percent.

Samples	Fungal Species	DNA	Sequence Length	Blast ID Number	Conidia Microscope Evaluation	Match Percent (%)
Plant Tissues	*B. bassiana*	CCGGTAAACCACAGTATCTACCTGATTCGAGGTCACGTTCAGAAGTTGGGTTTTTACGGCGTGGCCGCGTCGGGGTTCCGGTGCGAGCTGTATTACTGCGCAGAGGTCGCCGCGGACGGGCCGCCACTCCATTTCAGGGCCGGCGGTGTGCTGCCGGTCCCCAACGCCGACCTCCCCAAGGGGAGGTCGAGGGTTGAAATGACGCTCGA ACAGGCATGCCCGCCAGAATGCTGGCGGGCGCAATGTGCGTTCAAAGATTCGATGATTCACTGGATTCTGCAATTCACATTACTTATCGCGTTTCGCTGCGTTCTTCATCGATGCCAGAGCCAAGAGATCCGTTGTTGAAAGTTTTGATTCATTTGTTTTGCCTTGCGGCGTATTCAGAAGATGCTGGAATACAAGAGTTTGAGGTCCCCGGCGGGCCGCTGGTCCAGTCCGCGTCCGGGCTGGGGCGAGTCCGCCGAAGCAACGATAGGTAGGTTCACAGAAGGGTTAGGGAGTTGAAAACTCGGTAATGATCCCTCCGCTGGTTCACCAACGGAGACCTTGTTACGACTTTTACTTCCA	555	20200105GS1P2_E01_20	Yes	100
*T. confusum* baits	*B. bassiana*			20200105GS1P2_E02_20	Yes	100

**Table 2 jof-07-00142-t002:** Mean (±sd; *n* = 50) plant growth parameters of endophyte-free *V. vinifera* plants (control) and plants infested with *B. bassiana*. Means of the same row followed by the same letter are not significantly different (*p* < 0.05).

Plant Growth Parameters	Self-Rooted Control	Self-Rooted Endophtyic *B. bassiana*	R110 Control	R110 Endophtyic *B. bassiana*
Number Leaves	33.18 ± 3.71a	40.84 ± 2.9b	29.25 ± 5.77a	40.73 ± 3.15b
Height of plant	40.09 ± 1.10b	45.08 ± 2.69c	34 ± 1.96a	40.53 ± 1.80b
Thickness of the stem	10.63 ± 1.15a	11.58 ± 1.67a	11.08 ± 2.63a	11.84 ± 2.23a
Length of clematis	47.36 ± 8.62a	58.77 ± 19.15a	51.20 ± 10.83a	56.12 ± 17.66a
Length of Root	29.18 ± 3.83a	39 ± 2.56b	29.58 ± 4.55a	37.75 ± 1.58b
Leaf dry weight	19.04 ± 1.98a	22.43 ± 1.81a	20.69 ± 3.69a	25.18 ± 2.53a
Shoot dry weight	14 ± 6.14a	19.56 ± 6.52a	17.78 ± 10.31a	20.99 ± 8.46a
Stem dry weight	18.75 ± 4.59a	20.63 ± 6.16a	19.03 ± 8.72a	22.59 ± 12.99a
Root dry weight	35.02 ± 1.34a	57.63 ± 3.02b	49.48 ± 2.42a	81.15 ± 5.74c
Aboveground dry weight, A (g)	51.79	60.46	57.50	68.76
Belowground dry weight, B (g)	35.02	57.63	49.48	81.15
Ratio A/B	1.47	1.04	1.16	0.84
Total biomass (g)	86.81 ± 2.54a	118.09 ± 1.98c	106.98 ± 1.85b	149.91 ± 3.17d

**Table 3 jof-07-00142-t003:** *T. confusum* bait used to trap *B. bassiana*; mean percentage of mortality on vine soil and mean percentage of mycelium present on cadavers.

Soil Mortality of *T. confusum*				
Treatment	2d	4d	6d	Mycelium Present on Dead Adults
Self-rooted Control	3.67 ± 2.75a	6.13 ± 1.96a	7.89 ± 2.11a	0 ± 0a
Self-rooted Endophytic *B*. *bassiana*	44.06 ± 3.21b	63.54 ± 5.40b	91.09 ± 3.56b	87.97 ± 5.08b
R110 Control	4.54 ± 2.67a	6.75 ± 4.13a	8.32 ± 2.51a	0 ± 0a
R110 Endophytic *B. bassiana*	52.79 ± 4.67c	71.11 ± 0.42c	87.47 ± 1.96b	73.45 ± 3.51c

## Data Availability

The data presented in this study are available on request from the corresponding author.

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
