# Peer review of "Beauveria bassiana* Endophytic Strain as Plant Growth Promoter: The Case of the Grape Vine *Vitis vinifera"

_jof, 2021, doi:10.3390/jof7020142_

Round 1
Reviewer 1 Report
The authors have taken in count my comments to the previous version. I thank then, as the current version is much improved. I think this manuscript can be accepted after checking on some typos, grammar and misspellings on the text. I add the pdf version of the reviewed manuscript with my comments and highlights included.

Author Response
First, we would like to express our gratitude to the reviewers. Almost all their suggestions have been incorporated in the text and the MS has been greatly improved.
All the minor errors (syntax, rewriting, slight word modifications, poor structure, grammar etc) had been corrected.
Reviewer 2 Report
Please see the attached file for some suggestions, comments and edits.
I still have concerns of the ID of the B bassiana and you haven't provided sufficient information that the Bb you put on is the same as the Bb that was reisolated.
You need to look at more specific genes to do this, not just ITS which is ok for ID only- you need to be differentiating with B-tubulin or EFT1 etc

Author Response
First, we would like to express our gratitude to the reviewers. Almost all their suggestions have been incorporated in the text and the MS has been greatly improved.
All the minor errors (syntax, rewriting, slight word modifications, poor structure, grammar etc) had been corrected.
remove 'major characteristic'- add properties? ....however i'm not sure these properties differentiate it from Metarhizium. Can you add more details?
Reply: Thank you for your comment. In some cases the major characteristic are different from Metarhizium. Details added (lines 63-65)
this doesn;t make sense to me either, is it plant growth in response to colonisation? Be more clear and succinct
Reply: Plant growth is closely related to the colonization. The sentence has been rephrased (lines 79-81).
this needs rewriting, it doesn't flow. The experiment was set-up as a randomized block design ..then add the other info after rewriting I see no need for the photos here
Reply: Appropriate correction has been made. We believe that the photos are necessary to explain the experiment set up.
is this the aperture? put in brand of sieve
Reply: The sieve hasn’t had a brand
what volume containers?
Reply: The collected soil was placed in small plastic containers (2 x 3 cm).
How did you determine that you were assessing Beauveria and not some other fungi on the surface?
Reply: We performed morphological identification of conidia at 400x magnification
and the plates were set-up in a WHAT DESIGN
Reply: In randomized block design with ten Petri dishes per sample.
reduce the detail here. Just write something like ' genomic DNA was extracted from the pure cultures using the method of X. PCR using the universal primers ITS/ITS5 were used to amplify a region of the ITS spacer region... (don;t need to add primer detail).. These photos are pretty average. Get better photos. No scale bars, poor focus,
Reply: We believe that these details are crucial in this section.
did you get the expected amplicon size?
Reply: Yes. The molecular identification methodology is described in M&M.
what is conidia evaluation? The table does not give me any confidence that you have identified Bb
Reply: We mean the morphological identification of conidia at 400x magnification. Appropriate correction has been made. Thank you for your comment.
what have other studies found? e.g. different plants with Bb or Metarhizium spp.? any explantations offered elsewhere? Sheer proximity to where the Bb was applied?
Reply: We did not locate any study comparing the degree of colonization in various plant parts. Discussion about endophyte colonization in the grapevine and other crops are presented in the discussion.
I'm not of fan of claims to increase survivorship- more likely just grows more efficiently, which would lead to better yield of berries?yield? surely that could be useful
Reply: We totally agree. However, we cannot support this with the present study. However we know very well, from unpublished data, that the fungi can certainly increase the yield under certain circumstances.